biomechanics/robotics

ankle-foot prosthesis, uneven terrain, balance, transtibial amputation

**Author for correspondence:**
Steven H. Collins
e-mail: stevecollins@stanford.edu

# The effects of ground-irregularity-cancelling prosthesis control on balance over uneven surfaces

Vincent L. Chiu, Alexandra S. Voloshina and Steven H. Collins

Department of Mechanical Engineering, Stanford University, Stanford, CA 95014, USA

 VLC, 0000-0002-4265-7882; ASV, 0000-0002-6422-4666;
SHC, 0000-0002-3997-3374

Over half of individuals with a lower-limb amputation are unable to walk on uneven terrain. Using a prosthesis emulator system, we developed an irregularity-cancelling controller intended to reduce the effect of disturbances resulting from uneven surfaces. This controller functions by changing the neutral angles of two forefoot digits in response to local terrain heights. To isolate the effects of the controller, we also programmed a spring-like controller that maintained fixed neutral angles. Five participants with transtibial amputation walked on a treadmill with an uneven walking surface. Compared with the spring-like controller, the irregularity-cancelling controller reduced ankle torque variability by 41% in the sagittal plane and 64% in the frontal plane. However, user outcomes associated with balance were mostly unaffected; only trunk movement variability was reduced, whereas metabolic rate, mediolateral centre of mass motion, and variabilities in step width, step length and step time were unchanged. We conclude that reducing ankle torque variability of the affected limb is not sufficient for reducing the overall effect of disturbances due to uneven terrain. It is possible that other factors, such as changes in step height or disturbances to the intact limb, play a larger role in difficulty balancing while walking over uneven surfaces.

## 1. Introduction

Navigating uneven terrain, such as gravel, grass or trails, poses a challenge for users of lower-limb prosthetic devices. Approximately 60% of people with amputation cite an inability to walk on natural surfaces as a major limitation [1]. Walking on such surfaces leads to slower self-selected walking speeds, greater energy expenditure, and an increased fear of falling,

compared with unaffected individuals [2,3]. In turn, these limitations can lead to reduced mobility and overall quality of life.

When walking on uneven terrain, individuals without impairment show increased variability and range of motion in their ankle angle. This results in similar shank kinematics on uneven terrain compared with those observed during walking on level ground [4]. However, this strategy is not possible for people with a lower-limb amputation using passive prostheses, since such devices lack the ability to respond to changing surfaces. As such, users of passive prosthetic feet must compensate for uneven surfaces by modulating their centre of mass state using hip torque or changing their base of support with foot placement [5]. A prosthesis that could actively sense and compensate for uneven terrain might offload some control effort required from the user and reduce the magnitude of hip torque or foot placement adjustments.

Active ankle compensation is important in both sagittal and frontal planes. Active control in the sagittal plane has been the focus of many powered devices and allows for navigation of slopes and stairs as well as the ability to restore powered push-off [6,7]. However, active control in the frontal plane may be more important, as previous literature suggests that passive dynamic properties of the lower limbs are more stable in the sagittal plane than in the frontal plane [8]. In order for a prosthesis to fully compensate for uneven terrain, it should have control in both sagittal and frontal planes.

Prosthetic devices have implemented methods to reduce disturbances from walking on uneven terrain. For example, passive devices commonly adopt a split-toe design to increase compliance and to better conform to uneven terrain. Similarly, some devices use hydraulic ankles to allow for a wider range of ankle motion and to adapt to sloped surfaces [9]. However, both of these features cannot actively compensate for surface irregularities, so disturbance torques in both frontal and sagittal planes are still present. Some microprocessor-controlled feet use sensors and motorized elements to detect gait transitions and actively adapt between level ground, slopes and stairs [10,11]. By contrast, walking on uneven terrain does not involve discrete gait transitions yet still results in changing ground angles on each step. These devices are currently unable to detect and compensate for new angles within a single step.

Devices specifically aimed at adapting to uneven terrain are also in development. One method of adapting to new ground angles on each step is to mechanically switch between low and high ankle stiffnesses at critical points during stance [12]. Ankle stiffness is low during initial heel contact, which allows the device to conform to the slope of the ground. After a certain amount of weight is placed on the prosthesis, ankle stiffness increases to provide sufficient ankle torque for mid-stance support and late-stance push-off. This method effectively changes the neutral angle of the device based on the ground shape of each step. Another method is to command ankle torques in relation to shank angle using a series elastic actuator [4]. Commanding ankle torques based on shank angle allows the device to adapt to different ground slopes on each step without having to explicitly detect the angle of the slope. However, these methods have not been applied to ankle inversion–eversion to compensate for frontal plane disturbances. More importantly, the effects of these methods on balance-related biomechanical outcomes have not yet been evaluated.

Several biomechanical outcomes are associated with changes in balance during walking. For example, when compared with walking on level ground, walking on uneven terrain results in increased energy expenditure, shorter steps and increased foot placement variability [13]. Use of a prosthesis is associated with further changes in these balance-related outcomes. People with amputation exhibit greater variability in foot placement, such as step width variability [14–16], step length variability [16] and step time variability [16]. They also show greater variability in centre of mass motion, such as trunk movement variability [14] and mediolateral centre of mass motion [15]. These kinematic changes are a result of control strategies including changing the base of support with foot placement and modulating centre of mass state using hip torque. A prosthetic device that reduces balance-related control effort for the user is likely to show improvements in these biomechanical outcomes.

In this manuscript, we present an irregularity-cancelling controller developed to reduce the effect of disturbances due to uneven terrain. Similar to previously developed devices, this controller adapts to changes in ground angle on every step, but extends this functionality to the frontal plane. We tested the efficacy of this controller by comparing people walking on an uneven terrain treadmill with an active prosthesis using the irregularity-cancelling controller and a simpler, spring-like controller. The spring-like controller retains many of the same parameters as the irregularity-cancelling controller, but maintains a fixed behaviour on every step. We hypothesized that users walking with the irregularity-cancelling controller would experience reduced metabolic cost and kinematic variability compared with walking with the spring-like controller. We expect this study to contribute towards the

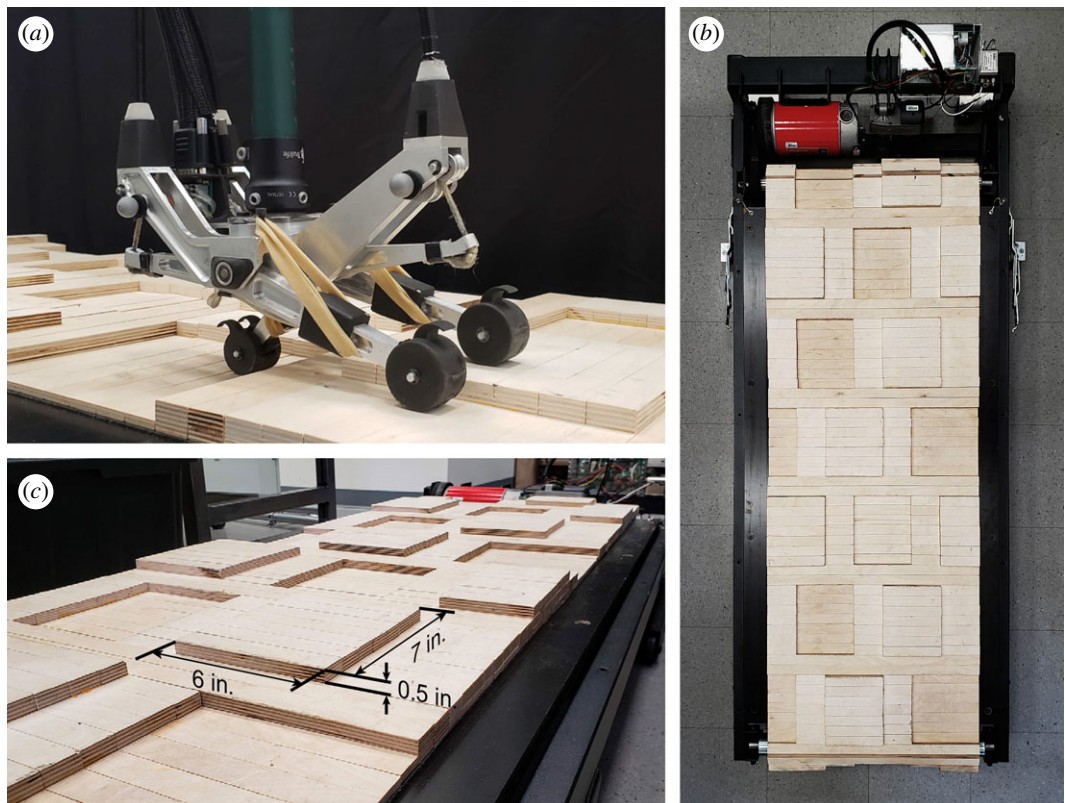

**Figure 1.** (*a*) Prosthesis emulator placed on the uneven terrain treadmill with a digit on each of the three terrain heights. (*b*) Top-down view of the uneven terrain treadmill. (*c*) Close-up of the terrain profile with dimensions.

development of active prostheses designed to improve balance on uneven terrain, and thereby quality of life, for people with lower-limb amputation.

## 2. Methods

We developed an irregularity-cancelling controller for a three degree-of-freedom prosthesis emulator [17] capable of controlling its three digits independently (figure 1*a*). Five participants with unilateral transtibial amputation walked on a custom-made uneven terrain treadmill that mimicked walking on an unpredictable natural surface, similar to a hiking trail. Participants walked on the terrain treadmill while using their prescribed prosthesis, the prosthesis emulator with the spring-like controller, and the prosthesis emulator with the irregularity-cancelling controller. We evaluated user preference, metabolic rate, mediolateral centre of mass motion, and variabilities of step width, step length, step time and trunk movement.

### 2.1. Uneven terrain treadmill

We constructed an uneven terrain treadmill, similar to [18], that allowed for continuous walking in the laboratory environment. Several candidate terrain profiles were simulated to determine the distribution of disturbances in frontal and sagittal planes. The selected terrain profile demonstrated the most uniform distribution of disturbances. It consisted of rectangles 18 cm (7 in) long, varying between 7.6 and 15 cm (3–6 in) in width, located at three distinct heights in 1.3 cm (0.5 in) increments (figure 1*b*). Additional information about the simulation can be found in the electronic supplementary material, text.

We designed the pattern of the terrain profile so it is particularly difficult for users to learn. In order for the pattern to repeat, users would need to have a step length of one-half, one-third, one-quarter, etc. of the treadmill belt length. Given the belt length of 320 cm, if the user walks at one-fifth that value (64 cm), then the pattern will repeat every 5 strides. Since the blocks that make up the terrain are 2.5 cm long, each step must have a length of 64 ± 1.3 cm. On average over the length of the trial, the user would need to walk with an average step length of nearly exactly 64 cm because consistently longer or shorter steps

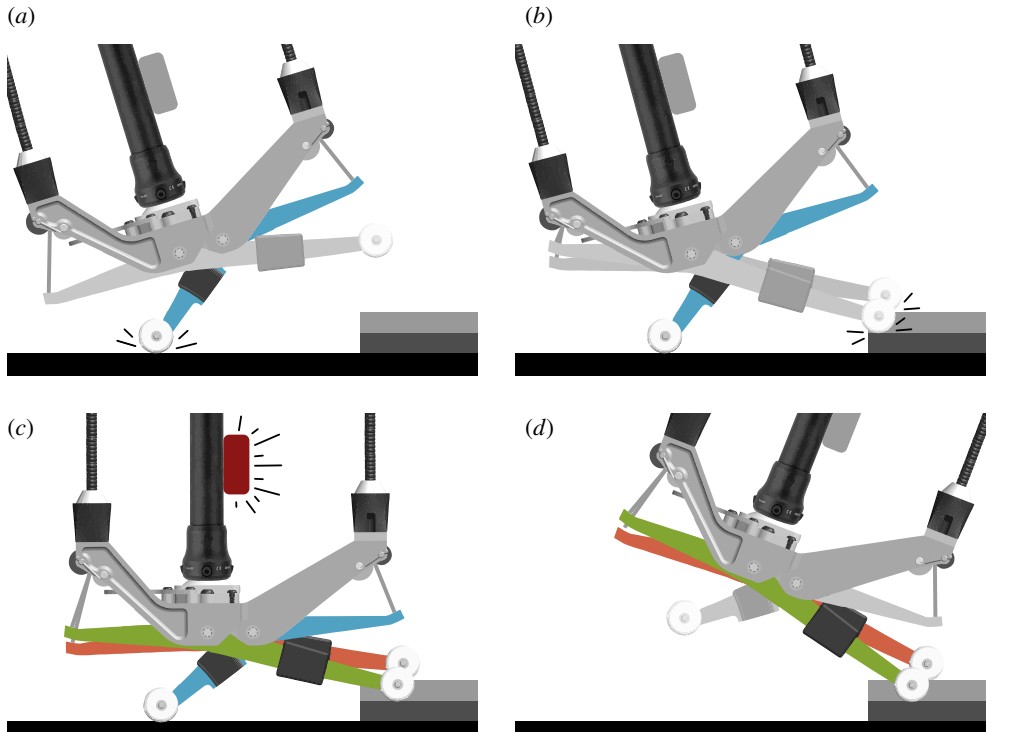

**Figure 2.** The three states of the irregularity-cancelling controller are Swing, Sense and Stance. (*a*) Heel strike triggers the transition from Swing to Sense. (*b*) The forefoot digits quickly touch down to gently sense the local terrain height during Sense. (*c*) An IMU on the shank triggers the transition from Sense to Stance at a preset angle and the forefoot digit angles are recorded. (*d*) During Stance, each forefoot digit emulates a virtual spring with the neutral angle set to the recorded angles.

would mean their step pattern would drift to adjacent terrain blocks, thus changing the pattern. To summarize, for a pattern to truly repeat, the user must walk at a specific step length with high accuracy and repeatability and memorize the corresponding pattern, all while being perturbed by uneven terrain. Anecdotally, this is more difficult than simply walking under the assumption that the terrain profile is random. To verify that participants did not use such a strategy, we analysed the pattern of foot angles during data collections and found it was not periodic (electronic supplementary material, text).

To fabricate the terrain surface, we attached wooden planks onto the belt of a commercially available exercise treadmill. The planks were cut from Baltic birch plywood and stapled to the belt from the underside. Each plank is 56 cm (22 in) wide to match the width of the treadmill belt and 2.5 cm (1 in) long to allow the belt and planks to move over the treadmill rollers. The varying heights in the terrain profile are achieved by gluing one or two additional planks onto the base plank. The crossbars on the treadmill frame were cut and replaced with custom-made beams to provide clearance for the wooden blocks. A document covering additional details on treadmill modification is provided in the electronic supplementary material.

## 2.2. Prosthesis controller

We implemented the irregularity-cancelling prosthesis controller on an existing tethered ankle-foot prosthesis emulator [17]. The emulator consists of a heel digit and two forefoot digits, all independently controlled. Each digit is equipped with an encoder and strain gauges to detect angular position and applied torques. These sensing capabilities, along with the wide range of motion, give the emulator the ability to conform to uneven terrain. We also attached an inertial measurement unit (IMU; Hillcrest Labs BNO080, Rockville, MD, USA), to the anterior aspect of the prosthesis pylon to measure shank angle in the laboratory reference frame [19].

The irregularity-cancelling controller is based on a state machine consisting of three states: Swing, Sense and Stance (figure 2). During the Swing state, the forefoot digits are slightly dorsiflexed to provide toe clearance. During the Sense state, the forefoot digits gently come into contact with the

ground in order to determine terrain height at each digit contact location. During the Stance state, each forefoot digit independently emulates a virtual spring, with the neutral angle adjusted to make the surface feel level, based on the terrain heights sensed during the previous Sense state. The heel digit emulates a virtual spring with a predefined neutral angle during all three states.

Transitions between states are triggered by gait events. The state machine transitions from the Swing state into the Sense state when the heel digit contacts the ground (figure 2a). During the Sense state, the forefoot digits plantarflex at a constant velocity until each forefoot digit makes contact with the ground. After each forefoot digit contacts the ground, it holds a torque of 3 N m, allowing it to maintain contact with the ground without resulting in significant plantarflexion torques (figure 2b). When the IMU registers that the transition shank angle has been reached, the angles of each forefoot digit are recorded and the state machine transitions to the Stance state (figure 2c). In this state, each forefoot digit independently emulates a virtual spring, where the neutral angle is set at the digit's respective recorded angle (figure 2d). The neutral angle, which changes on every step, allows for surface adaptation in both sagittal and frontal planes. The state machine cycles back to the Swing state when the forefoot digits leave the ground. The strain gauges on each digit are used to determine whether the digit is in contact with the ground. When torque on a digit rises above 2 N m, the controller registers that the digit has contacted the ground, and when torque falls below 2 N m, that digit has left the ground.

Non-variable parameters for the irregularity-cancelling controller, including heel and forefoot stiffnesses, plantarflexion velocity, IMU threshold angle and the heel neutral angle, were all set during a fitting session prior to testing. Heel and forefoot stiffnesses, as well as heel neutral angle, were set with guidance from the prosthetist. Plantarflexion velocity was set such that the forefoot digits would contact the ground before the shank angle crossed the threshold value. The IMU threshold angle was set by having the participant walk on a level treadmill and compare the spring-like and irregularity-cancelling controllers. The threshold angle was adjusted until the participant reported the irregularity-cancelling controller felt identical to the spring-like controller. Adjusting the IMU threshold angle is similar to changing the alignment of a prosthesis to be slightly plantarflexed or dorsiflexed. A table of these parameters can be found in the electronic supplementary material, text.

To evaluate the effects of the irregularity-cancelling controller relative to not compensating for the uneven surface, we also tested a spring-like controller. The spring-like controller functions similarly to the irregularity-cancelling controller but with the neutral angles of the forefoot digits fixed to user preference and not dependent on surface shape each step (figure 3).

## 2.3. Experimental design

Five healthy adults with unilateral transtibial amputation participated in the study, with an average age, height and body mass of $47 \pm 17$ years, $170 \pm 4.6$ cm and $66 \pm 6.5$ kg, respectively (table 1). Prior to testing, participants were scheduled for a fitting session in which a prosthetist fit them to the prosthesis emulator. Participants then acclimated to walking in the emulator on both level ground and uneven terrain. At the end of the session, we asked the participant to identify their fastest comfortable walking speed while walking on the uneven terrain. This speed was held constant for that participant for all trials of the experiment.

The study consisted of 3 days of training and 1 day of validation. Each training day consisted of one trial of walking on a level treadmill (Bertec, Columbus, OH, USA) with the participant's prescribed prosthesis, followed by five trials of walking on the uneven terrain treadmill with the assigned prosthesis condition for that day. The three conditions included the participant's prescribed device, the prosthesis emulator with the irregularity-cancelling controller, and the prosthesis emulator with the spring-like controller. The order of the conditions was randomized for each participant. All trials were 6 min long with a 5 min break in between. The protocol allowed participants to walk on each prosthesis condition for over 1000 steps, after which previous research has shown adaptations to prosthesis interventions occur at a slower rate [20]. The validation day consisted of walking with all three prosthesis conditions on the uneven terrain treadmill in a double-reversal protocol, where the three conditions were tested in a random order and then again in reverse.

Validating a controller designed to improve balance requires measurements of the user's balance. However, finding a true measure of an individual's level of balance while walking is still an open problem. In this manuscript, we report kinematic outcomes that have previously been shown to be significantly different between people with and without transtibial amputation when walking over uneven surfaces: step width variability [14–16], step length variability [16], step time variability [16],

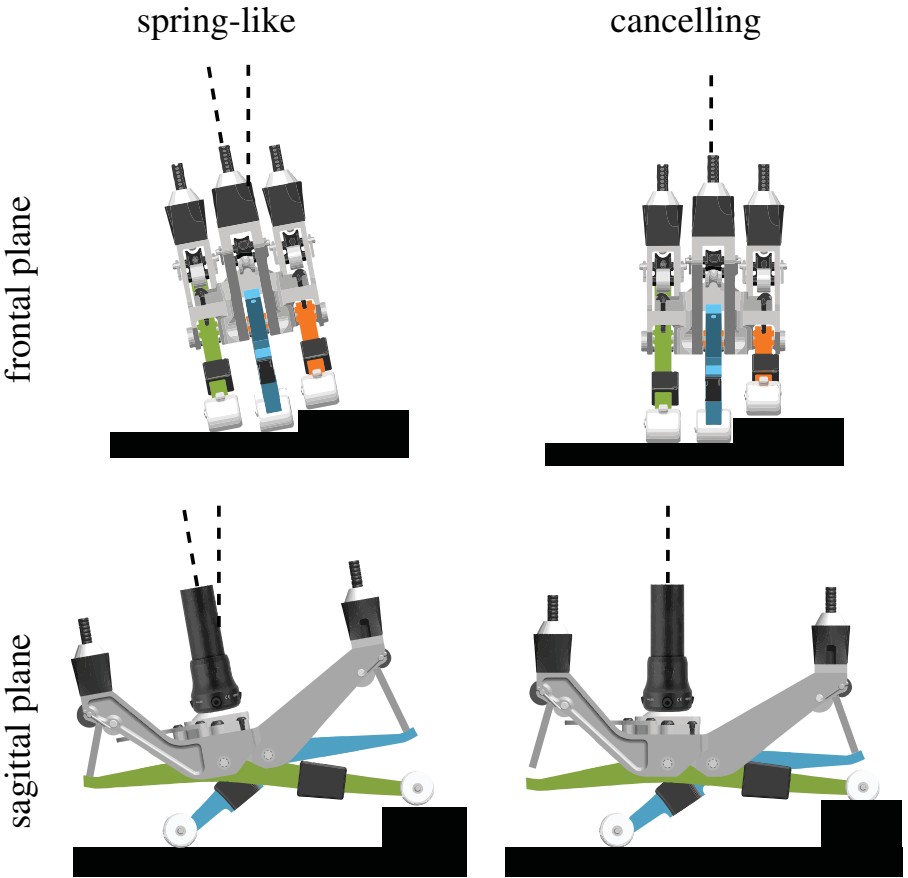

spring-like                    cancelling

frontal plane

sagittal plane

**Figure 3.** The irregularity-cancelling controller was designed to reduce disturbances due to uneven terrain in frontal and sagittal planes.

**Table 1.** Participant demographics.

| age | height (cm) | body mass (kg) | sex | years since amputation | type of amputation | prescribed device | walking speed (m s$^{-1}$) |
|---|---|---|---|---|---|---|---|
| 62 | 173 | 75 | M | 4 | vascular | Ossur Pro-Flex LP | 0.8 |
| 56 | 165 | 64 | M | 5 | traumatic | Ottobock Triton Harmony | 0.7 |
| 20 | 165 | 66 | F | 20 | congenital | Ossur Pro-Flex Pivot | 1.0 |
| 55 | 175 | 57 | M | 38 | traumatic | Ottobock Empower | 1.0 |
| 43 | 170 | 68 | M | 1 | traumatic | College Park Trustep | 1.0 |

trunk movement variability [14] and mediolateral centre of mass motion [15]. Additionally, we report metabolic rate, as it can be indicative of balance-related effort [13,21,22], and also perceived comfort, as it is one of the largest restrictions of prosthesis use [23].

We collected kinematic data using an eight-camera system at 100 Hz (Vicon Vero, Oxford, UK) that recorded the positions of 28 reflective markers placed on the participant at the feet, shanks, knees, hips, neck, shoulders, elbows and wrists. Step width, step length and step time were determined using the location of the heel marker at heel strike. Trunk movement variability was calculated as the mean standard deviation of the C7 position based on previous literature [14,24]. We estimated the centre of mass position using the location of the sacrum, previously shown to be highly correlated with actual centre of mass location [25]. We used a commercial respirometry system (Cosmed Quark CPET, Rome, Italy) to measure the rate of oxygen consumption and carbon dioxide production during walking and quiet standing conditions, then used these values to calculate metabolic rate using standard empirical

equations [26]. We calculated net metabolic rate by subtracting quiet standing metabolic cost from the metabolic cost of walking conditions.

## 2.4. Data processing and statistical analysis

To compare prosthesis behaviour between the irregularity-cancelling and spring-like controllers, we analysed the standard deviation of plantarflexion torque with respect to the shank angle progression in the sagittal plane during stance. Analysing ankle torque with respect to shank angle can show the magnitude of disturbances caused by uneven terrain and has been used to validate similar devices [4]. For each walking trial on the validation day involving the prosthesis emulator, we first calculated the standard deviation of plantarflexion torque across all steps then calculated a single RMS value. A larger RMS value is indicative of a greater effect of disturbances due to uneven terrain. Since the validation day followed a double-reversal protocol, we averaged the RMS values across the two repeated prosthesis conditions. We then averaged these RMS values across all participants to get values for the irregularity-cancelling and spring-like controllers. We repeated this process for ankle inversion-eversion torque with respect to the shank angle in the frontal plane.

Kinematic and metabolic data were analysed from the last 5 min of each walking trial on the validation day. Like the prosthesis data, we averaged the results across the repeated prosthesis conditions. We then normalized the results relative to walking on the level treadmill with their prescribed device. Participants completed this level-ground condition three times over the three training days and we used the average of those three conditions to define the baseline. At the end of the validation day, we asked participants to rank the comfort of the three prosthesis conditions from best to worst. Participants were not told which of the prosthesis emulator conditions was irregularity-cancelling and which was spring-like prior to ranking.

We performed a repeated-measures ANOVA to determine differences in user outcomes with respect to changes in prosthesis condition. We set the significance level at $\alpha = 0.05$ and performed Holm–Sidak multiple-comparison *post hoc* corrections for significant differences.

# 3. Results

We analysed prosthesis ankle torque variability in order to evaluate the effects of the irregularity-cancelling controller. We also analysed user outcome measures associated with balance to evaluate the efficacy of the irregularity-cancelling controller.

## 3.1. Device behaviour

The irregularity-cancelling controller, on average, resulted in a 41% reduction in RMS ankle torque variability in the sagittal plane and a 64% reduction in the frontal plane compared with the spring-like controller ($p < 0.001$ for both measures). Figure 4 shows representative data from one participant.

## 3.2. User outcomes

Walking with the irregularity-cancelling controller led to a 17% reduction in step width variability compared with walking with a prescribed prosthesis ($p = 0.018$). The irregularity-cancelling controller also led to a 15% reduction in trunk movement variability compared with walking in both the prescribed prosthesis ($p = 0.017$) and the spring-like controller ($p = 0.018$) (figure 5 and table 2). None of the other indicators of balance differed across conditions. We also measured average step width, length and time and found that they were not affected by prosthesis condition ($p > 0.119$), with values of $1.045 \pm 0.161$, $0.995 \pm 0.051$ and $1.046 \pm 0.060$, respectively. Results from participant surveys showed that four of the five participants ranked the irregularity-cancelling controller first or tied for first in subjective preference (table 3).

# 4. Discussion

We implemented spring-like and irregularity-cancelling controllers on a prosthesis emulator system. The irregularity-cancelling controller changes the neutral angle of each forefoot digit in response to local terrain height on each step, whereas the spring-like controller maintains a fixed neutral angle.

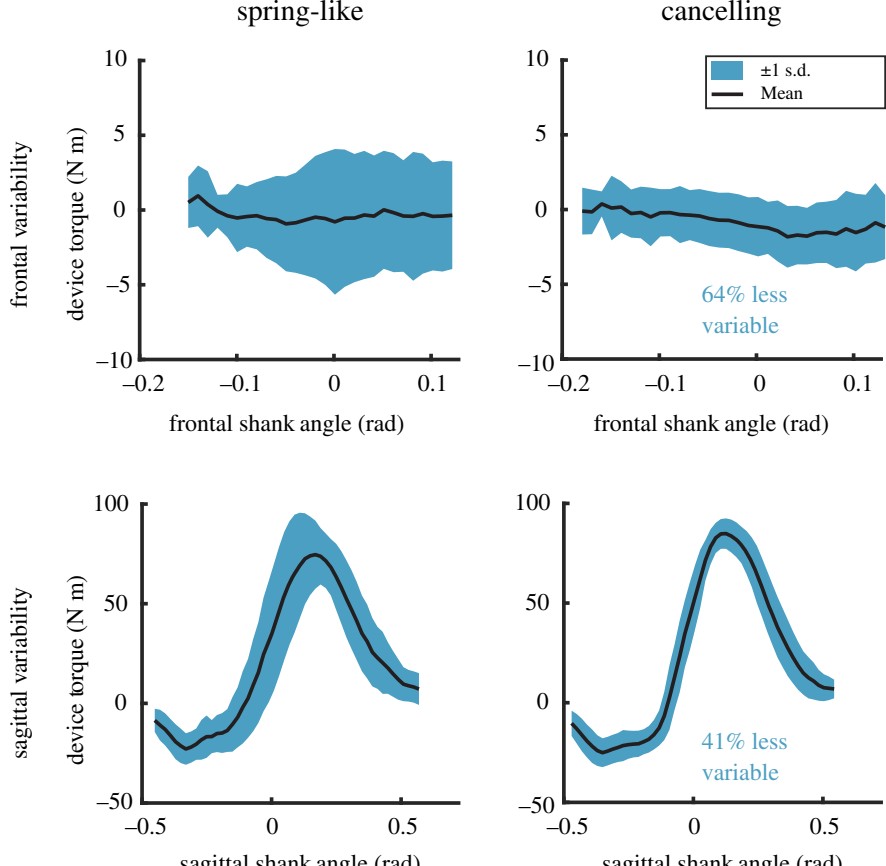

**Figure 4.** Sagittal and frontal ankle torque variability for one representative participant. On average, the irregularity-cancelling controller reduces RMS ankle torque variability compared with the spring-like controller by 41% in the sagittal plane and 64% in the frontal plane.

Although the irregularity-cancelling controller changed ankle kinetics as desired, whole-body kinematics were not as strongly affected. Overall, the irregularity-cancelling controller does not seem to be an effective method of reducing the effect of disturbances due to uneven terrain.

There are several potential explanations of why using the irregularity-cancelling controller did not result in the expected improvements in user outcomes despite significant reductions in ankle-torque variability. These reasons fall into the categories of limitations with the controller, interactions with the user and insufficient training.

One drawback of the irregularity-cancelling controller is that it is unable to compensate for differences in foot height at each step while walking on the terrain surface. In other words, although the irregularity-cancelling controller significantly reduces the effects of the slope at each step, the absolute height of each step may still differ between feet. Humans can adapt to differences in ground height using a combination of changes in posture at the ankle, knee and hip. However, this requires a much more complex control scheme than the simple one presented in this manuscript. It is possible that these height discrepancies on each step have a large influence on energy cost and kinematic variability when walking over uneven terrain. If this is the case, then the irregularity-cancelling controller is only compensating for a portion of the disturbances. The prosthesis emulator is capable of 1.3 cm of height adjustment, so future studies could compensate for changes in foot height to further mitigate the effects of uneven terrain. However, measuring the absolute position of the prosthesis with sub-centimetre resolution using fieldable sensors will be a technical challenge.

Interactions between the prosthesis emulator and the user are complex and another possible reason for the limited response in user outcomes. Since all the participants had a unilateral amputation, their intact limb was still subject to the disturbances due to uneven terrain. When walking over uneven terrain, people with a lower-limb amputation tend to rely more on their intact limb [3,27,28]. It is possible that results are dominated by disturbances to the contralateral limb and changes to the affected limb have a smaller influence on energy cost and kinematic variability. Another possibility is that users are affected by reduced sensory feedback on their affected limb. Reduced magnitude or

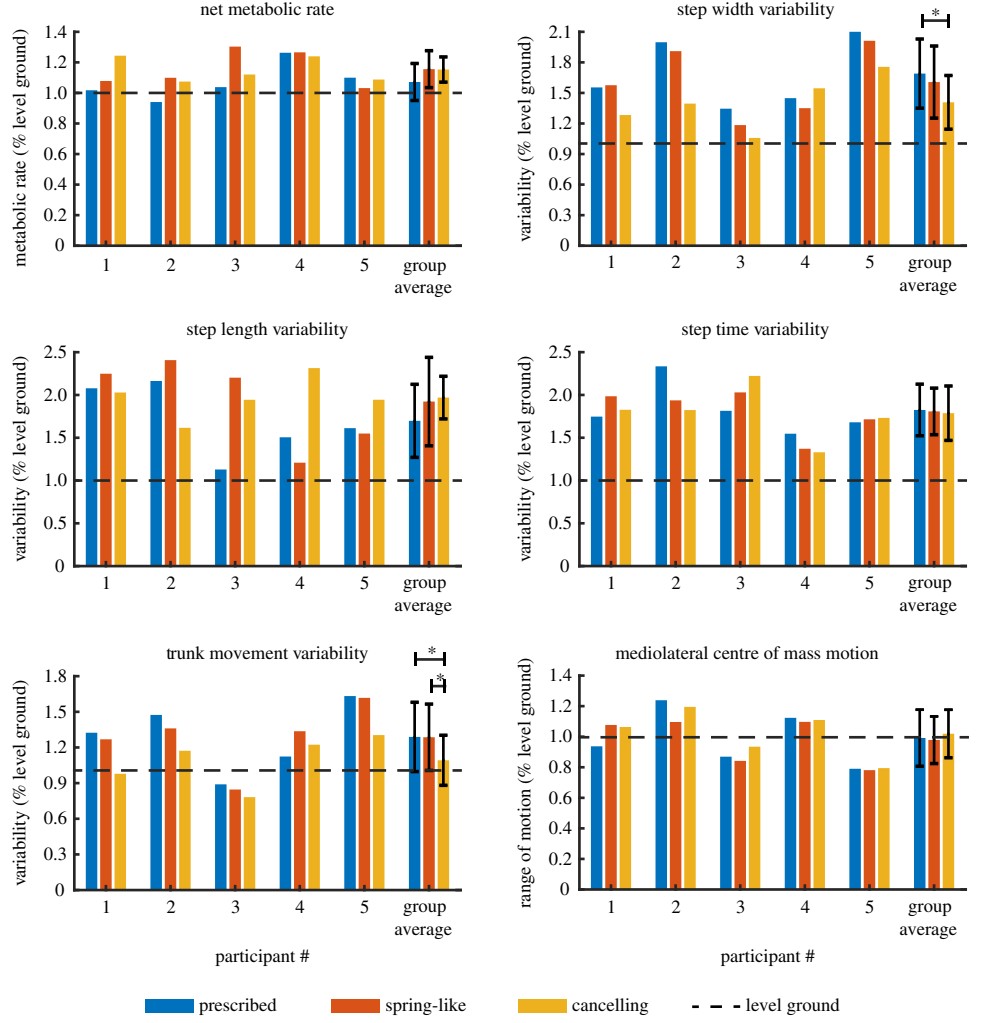

**Figure 5.** User outcomes while walking on uneven terrain for the three prosthesis conditions. Results are normalized to walking on a level treadmill using their prescribed device, indicated by the dashed line. *Statistical significance achieved ($p < 0.05$).

**Table 2.** Group-level means (and standard deviations) of user outcomes, normalized to level ground walking.

| dependent measure | prescribed (P) | spring-like (S) | cancelling (C) | *p*-value (ANOVA) | *p*-value (P/S) | *p*-value (P/C) | *p*-value (S/C) |
|---|---|---|---|---|---|---|---|
| metabolic rate | 1.072 (0.121) | 1.156 (0.121) | 1.153 (0.082) | 0.271 | 0.161 | 0.173 | 0.965 |
| step width variability | 1.690 (0.339) | 1.607 (0.354) | 1.408 (0.264) | *0.045* | 0.406 | *0.018* | 0.069 |
| step length variability | 1.698 (0.427) | 1.923 (0.517) | 1.970 (0.249) | 0.593 | 0.436 | 0.352 | 0.871 |
| step time variability | 1.825 (0.302) | 1.808 (0.272) | 1.787 (0.318) | 0.951 | 0.889 | 0.758 | 0.866 |
| trunk movement variability | 1.289 (0.292) | 1.286 (0.279) | 1.092 (0.211) | *0.026* | 0.965 | *0.017* | *0.018* |
| mediolateral centre of mass motion | 0.992 (0.185) | 0.979 (0.154) | 1.019 (0.157) | 0.504 | 0.710 | 0.441 | 0.265 |

Significant differences ($p < 0.05$) are in italics.

fidelity of sensory feedback could lead to uncertainty about the state of the prosthetic foot and may make it difficult for users to determine if disturbances are being cancelled. Socket fit can also affect balance during walking [29,30], with more compliant sockets leading to larger movements of the prosthesis

**Table 3.** Prosthesis condition preferences.

| participant | 1st | 2nd | 3rd |
| --- | --- | --- | --- |
| 1 | cancelling, spring-like | | prescribed |
| 2 | cancelling | spring-like | prescribed |
| 3 | spring-like | prescribed | cancelling |
| 4 | cancelling | spring-like | prescribed |
| 5 | cancelling, prescribed | | spring-like |

with respect to the residual limb during large torques produced by uneven terrain. However, we found that torque variability was significantly reduced with the irregularity-cancelling controller compared to the spring-like controller, suggesting that prosthesis behaviour, rather than socket behaviour, was dominant in this case. A fourth possibility is that each participant experiences different benefits from the irregularity-cancelling controller, depending on their priorities during walking. For example, Participant 1 showed reductions in step width and trunk movement variability, whereas Participant 3 only showed reductions in metabolic rate. If individuals indeed experience different benefits, then taking an average across the group washes out individual reductions. In other words, since people with amputation tend to exhibit highly variable gait patterns, it is possible that each participant draws their own benefit from irregularity-cancelling assistance.

Although the protocol was designed to allow for dedicated training days for each prosthesis condition, it is possible that the training time was insufficient. Previous studies have shown that people with an amputation adopt a more energetically costly but more stable gait using shorter, wider steps over challenging terrains [3,31]. In order to derive the most benefit from an irregularity-cancelling device, users would need to transition to a more normative gait which would, in turn, require increased reliance on balance assistance from the device. Unlearning a previous gait and learning to trust a new device may take significantly longer than the time permitted in this study. Intermittent training and frequent switching of prosthetic feet probably also hamper training effects, but are unavoidable limitations given our tethered, laboratory-based prosthesis emulator. These limitations would have to be addressed with an untethered device built specifically for navigating uneven terrain.

Survey results showed that four of the five participants ranked the irregularity-cancelling controller first or tied for first and three of the five participants ranked their prescribed prosthesis last. Some participants reported that walking on uneven terrain using their prescribed prostheses led to unpredictable perturbations, whereas the irregularity-cancelling controller felt more stable. The preference for the irregularity-cancelling controller over prescribed prostheses could be due to benefits that the controller provides but are not fully captured in the outcomes used in this study; there was no apparent correlation between survey results and user outcomes. However, since this was not a blinded study, the participants may have had an expectation bias towards the two prosthesis emulator conditions over their own prosthesis. Nevertheless, these survey results may indicate a benefit of prostheses designed to mitigate disturbances due to uneven terrain.

There are no trends in user outcomes that correlate with the masses of the participants' prescribed prostheses. Although we did not collect the masses of the devices used by the participants, we have provided estimates based on manufacturer data. We supplied shoes for the participants to wear during the experiment, with a mass of 250 g per shoe. The devices used by Participants 1 and 5 (Ossur Pro-flex LP and College Park Trustep) had a mass of about 650 g for the device and footshell. Adding the mass of the shoe results in a sum of 900 g. The devices used by Participants 2 and 3 (Ottobock Triton Harmony Ossur Pro-flex Pivot) had a mass of around 1150 g including footshell and shoe. The Ottobock Empower used by Participant 4 had a mass of around 2450 g including footshell and shoe. For comparison, the prosthesis emulator has a mass of about 1200 g and did not incorporate a footshell or shoe. Although Participant 5 reported that their prescribed foot felt lighter than the prosthesis emulator, we did not observe any trends in user outcomes that favoured lighter devices across any of our participants.

Since all measures were normalized to walking on a level treadmill while using their prescribed device, we can use data from the prescribed device condition to quantify how participants adapted their gait in response to uneven terrain (table 2). Specifically, we can see that terrain had a small or negligible effect on mediolateral centre of mass motion and metabolic rate (0.8% reduction and 7.2% increase, respectively), as well as mean step width, length and time (2.7% reduction, 0.5% reduction

and 2.7% increase, respectively). However, there was a large effect on step width, step length and step time variabilities (69%, 70% and 83% increase, respectively). This finding is consistent with previously reported data of individuals with amputation walking on terrain [3,32], and implies participants adapted to the terrain using foot placement strategies and relied less on centre of mass or torso adjustments. These adaptations are also similar to individuals without amputation, who experienced little to no changes in mean step parameters, but experienced significant increases in step width, length and time variability (35%, 23% and 27%, respectively) [13]. Notably, the increases in step variability were smaller and the increase in metabolic rate was greater than was reported in this study.

Participants were not given instructions on how to navigate the terrain or where to look while they walked. Although they were free to look down at the terrain, all participants looked forward during the experiment, and some mentioned that looking down at the terrain was disorienting. However, even if participants opted to look down, the nature of a treadmill makes it impossible to preview upcoming steps, save for the one about to be encountered, and participants would be little able to compensate for upcoming disturbances nor place their foot on easier sections of terrain. This suggests our results may not directly translate to overground terrain, where research has shown that people look ahead to plan their steps [33].

Natural walking surfaces often have several characteristics that could contribute to changes in gait dynamics, such as variation in terrain stiffness, traction or height. For this study, we specifically chose to focus on the effects of mitigating variations in terrain height. Future studies may explore methods of mitigating the other characteristics of uneven terrain, such as traction or stiffness.

## 5. Conclusion

In this study, we designed an irregularity-cancelling prosthesis controller intended to reduce the effect of disturbances due to uneven terrain. Compared with the spring-like controller, the irregularity-cancelling controller reduced ankle-torque variability in both frontal and sagittal planes and was ranked first in subjective preference by most participants. However, only trunk movement variability was reduced whereas metabolic rate, mediolateral centre of mass motion, and variabilities in step width, step length and step time were unchanged. It appears that simply reducing ankle torque variability of the affected limb is not sufficient for reducing the overall effect of disturbances due to uneven terrain. Future prosthetic devices aimed at improving balance when walking over uneven terrain will also want to consider additional modes of assistance, such as compensating for changes in step height, modulating centre of mass state, or assisting with foot placement.

Ethics. All participants provided written informed consent at the beginning of the study and were fitted to the end-effector by a licensed prosthetist. All experimental procedures were approved by the Stanford University Institutional Review Board under Protocol #44162.

Data accessibility. Raw and processed data, along with associated processing code, can be found at the Dryad Digital Repository: https://doi.org/10.5061/dryad.z8w9ghx8v [34].

Authors' contributions. V.L.C. carried out data collection and data analysis, participated in the design of the study and drafted the manuscript; A.S.V. participated in data collection, data analysis and design of the study and critically revised the manuscript; S.H.C. conceived of the study, coordinated the study and critically revised the manuscript. All authors gave final approval for publication and agree to be held accountable for the work performed therein.

Competing interests. We declare we have no competing interests.

Funding. This work was supported by the National Science Foundation (grant no. CBET-1511177).

Acknowledgements. The authors thank Susan Stenman, CP, for assistance with fitting of the device to our participants and Ashley Nguyen for assistance with data collection.

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
