## [Reviewer comments · Royal Society Open Science]

Review History

RSOS-201235.R0 (Original submission)

Review form: Reviewer 1

Is the manuscript scientifically sound in its present form?

Yes

Are the interpretations and conclusions justified by the results?

Yes

Is the language acceptable?

Yes

Do you have any ethical concerns with this paper?

No

Have you any concerns about statistical analyses in this paper?

No

Recommendation?

Accept with minor revision (please list in comments)

Comments to the Author(s)

Reducing Prosthetic ankle torque variability is insufficient for improving balance on uneven surfaces Review

Summary:

In this study the authors use a prosthesis emulator with 3 degrees of freedom (2 articulated toes and 1 articulated heel) to test a control method to reduce ankle torque variability on uneven surfaces. Passive prostheses used on uneven surfaces have no way of adapting to disturbances in the terrain and thus lead to high torque variability in the sagittal and frontal planes. This new controller is designed to adapt the spring equilibrium position with each step and reduce this torque variability. The authors hypothesize that with this controller, they will be able to improve balance metrics, comfort, and metabolic cost for the user. The controller is tested with 6 subjects walking on an uneven treadmill while using motion capture and VO₂ to record data on variability in step length, step width, step time, minimum margin of stability, trunk movement, and metabolic rate. The results from the adaptive controller is compared their prescribed prosthesis and to a spring like controller used on the same emulator. From this data set, the authors determine that the new controller reduces trunk mobility and step width variability but doesn't affect the other metrics of balance / stability.

The authors present their study well with sound science and well thought out analysis. They present the results in a way that is easy to follow and point out where their study could be extended. Below are a list of questions and minor concerns that came up while reviewing the study:

Page 1

52: delete "still," it seems like this was left over from a previous version where a different statement preceded it.

Page 2

33-34: I think you are referring to just the time that the prosthesis is on the ground but the sentence makes a broader assertion. Stolyarov 2020 demonstrates a controller that can adapt different terrain types within a single step.

Page 5

10-12: It would be nice to show some data to support the idea that no pattern was found. This could be done by demonstrating that the foot angle or height is aperiodic.

12-15: This simulation could be explained in more detail. Probably something for the supplementary but it might be good to have a figure in the main doc that helps the reader understand the design of the treadmill and trust that the repeating pattern is not easily learned.

41-44: This explanation is a bit hard to follow. Maybe the controller torques could be added to figure 2 to help the reader ground the control output to the physical position of the device. A torque / shank angle loop could also be very informative.

Page 8

Table 2: Is there any correlation between these responses and the empirical data? Also, there isn't any reason to abbreviate "prescribed" here.

Page 9

Table 3: replace "Adaptive" with "Cancelling" since that is how it is referred to throughout the rest of the paper. It also might be good to mark each of the conditions with their one letter abbreviation to help connect the first 3 data columns to the last 3. Also, why was the pairwise

statistical comparison only done on significant results? I guess it's because from the ANOVA we already know those numbers won't be significant, but as a reader, I still wanted to see the comparison.

Page 10

Figure 5: Because all of this data is normalized to the level walking condition, the "1" should be kept at a consistent height on each of the plots or it should be highlighted in some manner. As it is currently, it is hard to compare the 3 conditions to the level ground case. Also, the statistically significant * is a bit hard to understand on the trunk mobility plot. P-S-C are significantly different? P-S are significantly different despite looking the same?

Page 11

7-14: Biological feet don't change height either. Could we improve the stability of the user by doing something so different from biology? Would that affect the contralateral limb?

15-29: The prosthesis socket can also be a source of instability that may dominate the results. Was there any thought for the socket fit when selecting subjects for this study?

47-49: Thanks for mentioning this as a potential source of bias.

Page 12

8-13: Why wouldn't the subjects look down? Were they instructed not to? When walking on rough terrain, humans tend to look down and as the terrain gets rougher they look ahead fewer steps (Matthis 2018). Can we be certain that the subjects are using a natural gait when they are knowingly walking on an unpredictable surface and actively not watching where they are stepping?

Data:

Looking through your provided code, some files suggest 9 total participants in the study. Is this accurate? And if so, why were the others excluded from the analysis?

Review form: Reviewer 2

Is the manuscript scientifically sound in its present form?

Yes

Are the interpretations and conclusions justified by the results?

Yes

Is the language acceptable?

Yes

Do you have any ethical concerns with this paper?

No

Have you any concerns about statistical analyses in this paper?

No

Recommendation?

Major revision is needed (please make suggestions in comments)

Comments to the Author(s)

In this paper the authors present a novel controller for an ankle-foot prosthesis designed to help the amputee cope more easily with uneven terrain, as they might experience in daily life. The

controller is tested using a prosthetic emulator system, compared to a different controller, and the participants' regular prosthesis. Testing was conducted on an adapted treadmill with an uneven surface atop the treadmill belt. Various outcome measures are used to assess the effects of the controller. The biomechanical outcomes are focussed on the variability of that metric and how it is affected by the prosthetic controller. The work is highly novel, relevant and interesting. The paper is clearly written. The nature of the work strikes me as more exploratory in nature, rather than a complete piece of experimental work, but is still suitable for the remit of this journal. Below are my specific comments.

As stated above the work is somewhat exploratory, and therefore I think the title of the paper is too firmly stated. While I appreciate that the nature of this work does not lend itself to large cohorts, with only 5 participants, each of which is likely to have an individual response to the intervention, the study lacks the power to draw such firm conclusions. This compounded by the other limitations that the authors acknowledge, such as a limited familiarisation to the intervention. The title suggests that a clear result has been reached, and that the study disproves any value in the control strategy. The paper itself does not carry this tone, and I believe the title should be changed to reflect more closely the nature of the work/findings.

It would improve the paper if the authors could justify their main outcome metrics in the methods. Intuitively, one might think that reduced variability is a sign of improved stability, but this is not always the case in the execution of motor tasks. It could be that it is not about preventing the variability induced by the terrain, but the individual's ability to respond to it. Therefore, is variability relative to level ground the best metric to assess device efficacy? Some justification in the paper would be beneficial.

What were the masses of the different prosthetics and the emulator attachment used in the study? Mass could easily have a significant impact on metabolic power during walking, so this should be included and discussed briefly.

Several control parameters were fixed based on the 'user preference'. It would be useful to include a brief description of how user preference was established and what some of these parameters were (maybe in an appendix). Particularly, the threshold angle at which the device switched to 'stance' control seems relevant.

Some discussion of the results in comparison to how non-amputees respond to similar uneven terrain situations seems relevant. Results are presented relative to level ground, so are they similar adaptations to those made by non-amputees? (e.g. is an increase in step width expected?). Also in the context of the control scheme - do non-amputees seek to minimise ankle torque variability on uneven terrain, or is a varied response required to account for different perturbations?

I presume treadmill speed was the same for level ground and uneven terrain. Maybe the natural response would be to walk more slowly on uneven ground?

Decision letter (RSOS-201235.R0)

Dear Dr Chiu

The Editors assigned to your paper RSOS-201235 "Reducing prosthetic ankle torque variability is insufficient for improving balance on uneven surfaces" have now received comments from reviewers and would like you to revise the paper in accordance with the reviewer comments and any comments from the Editors. Please note this decision does not guarantee eventual acceptance.

Please submit your revised manuscript and required files (see below) no later than 21 days from today's (ie 20-Aug-2020) date. Note: the ScholarOne system will 'lock' if submission of the revision is attempted 21 or more days after the deadline. If you do not think you will be able to meet this deadline please contact the editorial office immediately.

Royal Society Open Science
openscience@royalsociety.org
on behalf of Dr Manoj Srinivasan (Associate Editor) and R. Kerry Rowe (Subject Editor)
openscience@royalsociety.org

Associate Editor Comments to Author (Dr Manoj Srinivasan):

Comments to the Author:

The reviewers provide a number of comments that are, as they say, readily addressable or mostly minor. One reviewer suggests revising the title of the article to make it clear that this was exploratory, given the lower subject numbers. We look forward to a revised submission that addresses the reviewer remarks.

Reviewer comments to Author:

Reviewer: 1

Comments to the Author(s)

Reducing Prosthetic ankle torque variability is insufficient for improving balance on uneven surfaces Review

Summary:

In this study the authors use a prosthesis emulator with 3 degrees of freedom (2 articulated toes and 1 articulated heel) to test a control method to reduce ankle torque variability on uneven surfaces. Passive prostheses used on uneven surfaces have no way of adapting to disturbances in the terrain and thus lead to high torque variability in the sagittal and frontal planes. This new controller is designed to adapt the spring equilibrium position with each step and reduce this torque variability. The authors hypothesize that with this controller, they will be able to improve balance metrics, comfort, and metabolic cost for the user. The controller is tested with 6 subjects walking on an uneven treadmill while using motion capture and VO₂ to record data on variability in step length, step width, step time, minimum margin of stability, trunk movement, and metabolic rate. The results from the adaptive controller is compared their prescribed prosthesis and to a spring like controller used on the same emulator. From this data set, the authors determine that the new controller reduces trunk mobility and step width variability but doesn't affect the other metrics of balance / stability.

The authors present their study well with sound science and well thought out analysis. They present the results in a way that is easy to follow and point out where their study could be extended. Below are a list of questions and minor concerns that came up while reviewing the study:

Page 1

52: delete "still," it seems like this was left over from a previous version where a different statement preceded it.

Page 2

33-34: I think you are referring to just the time that the prosthesis is on the ground but the sentence makes a broader assertion. Stolyarov 2020 demonstrates a controller that can adapt different terrain types within a single step.

Page 5

10-12: It would be nice to show some data to support the idea that no pattern was found. This could be done by demonstrating that the foot angle or height is aperiodic.

12-15: This simulation could be explained in more detail. Probably something for the supplementary but it might be good to have a figure in the main doc that helps the reader understand the design of the treadmill and trust that the repeating pattern is not easily learned.

41-44: This explanation is a bit hard to follow. Maybe the controller torques could be added to figure 2 to help the reader ground the control output to the physical position of the device. A torque / shank angle loop could also be very informative.

Page 8

Table 2: Is there any correlation between these responses and the empirical data? Also, there isn't any reason to abbreviate "prescribed" here.

Page 9

Table 3: replace "Adaptive" with "Cancelling" since that is how it is referred to throughout the rest of the paper. It also might be good to mark each of the conditions with their one letter abbreviation to help connect the first 3 data columns to the last 3. Also, why was the pairwise statistical comparison only done on significant results? I guess it's because from the ANOVA we already know those numbers won't be significant, but as a reader, I still wanted to see the comparison.

Page 10

Figure 5: Because all of this data is normalized to the level walking condition, the "1" should be kept at a consistent height on each of the plots or it should be highlighted in some manner. As it

is currently, it is hard to compare the 3 conditions to the level ground case. Also, the statistically significant * is a bit hard to understand on the trunk mobility plot. P-S-C are significantly different? P-S are significantly different despite looking the same?

Page 11

7-14: Biological feet don't change height either. Could we improve the stability of the user by doing something so different from biology? Would that affect the contralateral limb?

15-29: The prosthesis socket can also be a source of instability that may dominate the results. Was there any thought for the socket fit when selecting subjects for this study?

47-49: Thanks for mentioning this as a potential source of bias.

Page 12

8-13: Why wouldn't the subjects look down? Were they instructed not to? When walking on rough terrain, humans tend to look down and as the terrain gets rougher they look ahead fewer steps (Matthis 2018). Can we be certain that the subjects are using a natural gait when they are knowingly walking on an unpredictable surface and actively not watching where they are stepping?

Data:

Looking through your provided code, some files suggest 9 total participants in the study. Is this accurate? And if so, why were the others excluded from the analysis?

Reviewer: 2

Comments to the Author(s)

In this paper the authors present a novel controller for an ankle-foot prosthesis designed to help the amputee cope more easily with uneven terrain, as they might experience in daily life. The controller is tested using a prosthetic emulator system, compared to a different controller, and the participants' regular prosthesis. Testing was conducted on an adapted treadmill with an uneven surface atop the treadmill belt. Various outcome measures are used to assess the effects of the controller. The biomechanical outcomes are focussed on the variability of that metric and how it is affected by the prosthetic controller. The work is highly novel, relevant and interesting. The paper is clearly written. The nature of the work strikes me as more exploratory in nature, rather than a complete piece of experimental work, but is still suitable for the remit of this journal. Below are my specific comments.

As stated above the work is somewhat exploratory, and therefore I think the title of the paper is too firmly stated. While I appreciate that the nature of this work does not lend itself to large cohorts, with only 5 participants, each of which is likely to have an individual response to the intervention, the study lacks the power to draw such firm conclusions. This compounded by the other limitations that the authors acknowledge, such as a limited familiarisation to the intervention. The title suggests that a clear result has been reached, and that the study disproves any value in the control strategy. The paper itself does not carry this tone, and I believe the title should be changed to reflect more closely the nature of the work/findings.

It would improve the paper if the authors could justify their main outcome metrics in the methods. Intuitively, one might think that reduced variability is a sign of improved stability, but this is not always the case in the execution of motor tasks. It could be that it is not about preventing the variability induced by the terrain, but the individual's ability to respond to it. Therefore, is variability relative to level ground the best metric to assess device efficacy? Some justification in the paper would be beneficial.

What were the masses of the different prosthetics and the emulator attachment used in the study? Mass could easily have a significant impact on metabolic power during walking, so this should be included and discussed briefly.

Several control parameters were fixed based on the 'user preference'. It would be useful to include a brief description of how user preference was established and what some of these parameters were (maybe in an appendix). Particularly, the threshold angle at which the device switched to 'stance' control seems relevant.

Some discussion of the results in comparison to how non-amputees respond to similar uneven terrain situations seems relevant. Results are presented relative to level ground, so are they similar adaptations to those made by non-amputees? (e.g. is an increase in step width expected?). Also in the context of the control scheme – do non-amputees seek to minimise ankle torque variability on uneven terrain, or is a varied response required to account for different perturbations?

I presume treadmill speed was the same for level ground and uneven terrain. Maybe the natural response would be to walk more slowly on uneven ground?

===PREPARING YOUR MANUSCRIPT===

===PREPARING YOUR REVISION IN SCHOLARONE===

Author's Response to Decision Letter for (RSOS-201235.R0)

See Appendix A.

RSOS-201235.R1 (Revision)

Review form: Reviewer 1

Is the manuscript scientifically sound in its present form?

Yes

Are the interpretations and conclusions justified by the results?

Yes

Is the language acceptable?

Yes

Do you have any ethical concerns with this paper?

No

Have you any concerns about statistical analyses in this paper?

No

Recommendation?

Accept as is

Comments to the Author(s)

The authors have improved on their manuscript and answered all of the reviewers' questions. The additions elevate the understanding of the study and clear up misunderstandings from the previous version. I am excited to see how this research develops in the future.

Review form: Reviewer 2

Is the manuscript scientifically sound in its present form?

Yes

Are the interpretations and conclusions justified by the results?

Yes

Is the language acceptable?

Yes

Do you have any ethical concerns with this paper?

No

Have you any concerns about statistical analyses in this paper?

No

Recommendation?

Accept as is

Comments to the Author(s)

Thank you to the authors for acknowledging the review comments and making comprehensive additions and changes. I am happy that my concerns were addressed.

Decision letter (RSOS-201235.R1)

Dear Dr Chiu,

It is a pleasure to accept your manuscript entitled "The effects of ground-irregularity-cancelling prosthesis control on balance over uneven surfaces" in its current form for publication in Royal Society Open Science. The comments of the reviewer(s) who reviewed your manuscript are included at the foot of this letter.

Kind regards,

Andrew Dunn

on behalf of Dr Manoj Srinivasan (Associate Editor) and R. Kerry Rowe (Subject Editor)

Reviewer comments to Author:

Reviewer: 1

Comments to the Author(s)

The authors have improved on their manuscript and answered all of the reviewers' questions. The additions elevate the understanding of the study and clear up misunderstandings from the previous version. I am excited to see how this research develops in the future.

Reviewer: 2

Comments to the Author(s)

Thank you to the authors for acknowledging the review comments and making comprehensive additions and changes. I am happy that my concerns were addressed.

Appendix A

Reviewer: 1

Comments to the Author(s)

Reducing Prosthetic ankle torque variability is insufficient for improving balance on uneven surfaces Review

Summary:

In this study the authors use a prosthesis emulator with 3 degrees of freedom (2 articulated toes and 1 articulated heel) to test a control method to reduce ankle torque variability on uneven surfaces. Passive prostheses used on uneven surfaces have no way of adapting to disturbances in the terrain and thus lead to high torque variability in the sagittal and frontal planes. This new controller is designed to adapt the spring equilibrium position with each step and reduce this torque variability. The authors hypothesize that with this controller, they will be able to improve balance metrics, comfort, and metabolic cost for the user. The controller is tested with 6 subjects walking on an uneven treadmill while using motion capture and VO₂ to record data on variability in step length, step width, step time, minimum margin of stability, trunk movement, and metabolic rate. The results from the adaptive controller is compared their prescribed prosthesis and to a spring like controller used on the same emulator. From this data set, the authors determine that the new controller reduces trunk mobility and step width variability but doesn't affect the other metrics of balance / stability.

The authors present their study well with sound science and well thought out analysis. They present the results in a way that is easy to follow and point out where their study could be extended. Below are a list of questions and minor concerns that came up while reviewing the study:

Thank you for your thoughtful comments.

Page 1

52: delete "still," it seems like this was left over from a previous version where a different statement preceded it.

We have incorporated this suggestion. The sentence now reads:

Navigating uneven terrain, such as gravel, grass, or trails, poses a challenge for users of lower-limb prosthetic devices.

Page 2

33-34: I think you are referring to just the time that the prosthesis is on the ground but the sentence makes a broader assertion. Stolyarov 2020 demonstrates a controller that can adapt different terrain types within a single step.

There have been several publications, including Stolyarov 2020, that demonstrate impressive methods to predict gait transitions, typically between level ground, slopes, and/or stairs. However, we believe this to be a different problem from that of walking solely on uneven ground where there are no distinct gait transitions, yet each step may be different from the last. We have clarified this point in the main text.

Prosthetic devices have implemented methods to reduce disturbances from walking on uneven terrain. For example, passive devices commonly adopt a split-toe design to increase compliance and to better conform to uneven terrain. Similarly, some devices use hydraulic ankles to allow for a wider range of ankle motion and to adapt to sloped surfaces [Ernst 2020]. However, both of these features cannot actively compensate for surface irregularities, so disturbance torques in both frontal and sagittal planes are still present. **Some** microprocessor-controlled feet utilize sensors and motorized elements to **detect gait transitions and actively adapt between level ground, slopes, and stairs** [Fradet 2010, Stolyarov2020]. **In contrast, walking on uneven terrain does not involve discrete gait transitions yet still results in changing ground angles on each step. These devices** are currently unable to detect and compensate for new angles within a single step.

Page 5

10-12: It would be nice to show some data to support the idea that no pattern was found. This could be done by demonstrating that the foot angle or height is aperiodic.

We now include these data as supplementary text, and refer to them in the corresponding point in the main text.

We designed the pattern of the terrain profile so it is particularly difficult for users to learn. In order for the pattern to repeat, users would need to have a step length of one-half, one-third, one-quarter, etc. of the treadmill belt length. Given the belt length of 320 cm, if the user walks at one-fifth that value (64 cm), then the pattern will repeat every 5 strides. Since the blocks that make up the terrain are 2.5 cm long, each step must have a length of 64 +/- 1.3 cm. On average over the length of the trial, the user would need to walk with an average step length of nearly exactly 64 cm because consistently longer or shorter steps would mean their step pattern would drift to adjacent terrain blocks, thus changing the pattern. To summarize, for a pattern to truly repeat, the user must walk at a specific step length with high accuracy and repeatability and memorize the corresponding pattern, all while being perturbed by uneven terrain. Anecdotally, this is more difficult than simply walking under the assumption that the terrain profile is random. **To verify that participants did not use such a strategy, we analyzed the pattern of foot angles during data collections and found it was not periodic (Supplementary Text).**

Supplementary Text:

To demonstrate the terrain pattern is aperiodic, we calculated the sagittal ankle angle when the shank was vertical on each stride. Fig. 2 shows a representative trial of a participant walking on uneven terrain using the irregularity-cancelling controller. We do not have corresponding data for level ground since the participants only used their prescribed prosthesis for those trials.

Foot angle when shank is vertical

Figure 2: A representative walking trial shows the terrain results in aperiodic foot angle

12-15: This simulation could be explained in more detail. Probably something for the supplementary but it might be good to have a figure in the main doc that helps the reader understand the design of the treadmill and trust that the repeating pattern is not easily learned. We have provided additional information about the simulation in Supplementary Text and expanded on why this repeating pattern is not easily learned in the Methods section.

Supplementary Text:

The unraveled treadmill belt measures 320 cm (126 in) long and 56 cm (22 in) wide. We discretized the terrain pattern into a 126x22 matrix with the value at each index equal to the height of the treadmill at that location. We then evaluated the sagittal and frontal angles of the prosthesis at every location in the matrix to form a histogram of disturbance angles for that terrain profile.

Fig. 1 shows four candidate patterns and their corresponding histograms. The Random profile consists of random heights at each index. This profile does not have a very uniform disturbance distribution and is extremely difficult to fabricate due to its complexity. The Random Blocks

profile uses random heights in larger blocks and results in more uniform disturbances. The Stripes profile uses a repeating block pattern which improves ease of fabrication due to repeating structures, but does not have a very uniform disturbance distribution. The Waves profile offers a good balance between disturbance distribution and ease of fabrication and was the profile we chose to build.

Figure 1: Four candidate terrain profiles and associated disturbance angle histograms

Methods:

We designed the pattern of the terrain profile so it is particularly difficult for users to learn. In order for the pattern to repeat, users would need to have a step length of one-half, one-third, one-quarter, etc. of the treadmill belt length. Given the belt length of 320 cm, if the user walks at

one-fifth that value (64 cm), then the pattern will repeat every 5 strides. Since the blocks that make up the terrain are 2.5 cm long, each step must have a length of 64 ± 1.3 cm. On average over the length of the trial, the user would need to walk with an average step length of nearly exactly 64 cm because consistently longer or shorter steps would mean their step pattern would drift to adjacent terrain blocks, thus changing the pattern. To summarize, for a pattern to truly repeat, the user must walk at a specific step length with high accuracy and repeatability and memorize the corresponding pattern, all while being perturbed by uneven terrain. Anecdotally, this is more difficult than simply walking under the assumption that the terrain profile is random. To verify that participants did not use such a strategy, we analyzed the pattern of foot angles during data collections and found it was not periodic (Supplementary Text).

41-44: This explanation is a bit hard to follow. Maybe the controller torques could be added to figure 2 to help the reader ground the control output to the physical position of the device. A torque / shank angle loop could also be very informative.

We are concerned that including a torque/shank angle loop in this section of the manuscript could give readers the impression that we are explicitly controlling this relationship, which is not the case. However, we acknowledge that the explanation could be expanded upon to be more clear and have done so in the main text.

The irregularity-cancelling controller is based on a state machine consisting of three states: Swing, Sense, and Stance (Fig. 2). During the Swing state, the forefoot digits are slightly dorsiflexed to provide toe clearance. During the Sense state, the forefoot digits gently come into contact with the ground in order to determine terrain height at each digit contact location. During the Stance state, each forefoot digit independently emulates a virtual spring, with the neutral angle adjusted to make the surface feel level, based on the terrain heights sensed during the previous Sense state. The heel digit emulates a virtual spring with a predefined neutral angle during all three states.

Transitions between states are triggered by gait events. The state machine transitions from the Swing state into the Sense state when the heel digit contacts the ground (Fig. 2a). During the Sense state, the forefoot digits plantarflex at a constant velocity until each forefoot digit makes contact with the ground. After each forefoot digit contacts the ground, it holds a torque of 3 Nm, allowing it to maintain contact with the ground without resulting in significant plantarflexion torques (Fig. 2b). When the IMU registers that the transition shank angle has been reached, the angles of each forefoot digit are recorded and the state machine transitions to the Stance state (Fig. 2c). In this state, each forefoot digit independently emulates a virtual spring, where the neutral angle is set at the digit's respective recorded angle (Fig 2d). The neutral angle, which changes on every step, allows for surface adaptation in both sagittal and frontal planes. The state machine cycles back to the Swing state when the forefoot digits leave the ground. The strain gauges on each digit are used to determine whether the digit is in contact with the ground. When torque on a digit rises above 2 Nm, the controller registers that the digit has contacted the ground, and when torque falls below 2 Nm, that digit has left the ground.

Table 2: Is there any correlation between these responses and the empirical data? Also, there isn't any reason to abbreviate "prescribed" here.

We do not see any correlations and have updated the paragraph in the Discussion section. We have also removed the abbreviation of "Prescribed" to improve readability.

Survey results showed that four of the five participants ranked the irregularity-cancelling controller first or tied for first and three of the five participants ranked their prescribed prosthesis last. Some participants reported that walking on uneven terrain using their prescribed prostheses led to unpredictable perturbations, whereas the irregularity-cancelling controller felt more stable. The preference for the irregularity-cancelling controller over prescribed prostheses could be due to benefits that the controller provides but are not fully captured in the outcomes used in this study; **there was no apparent correlation between survey results and user outcomes**. However, since this was not a blinded study, the participants may have had an expectation bias towards the two prosthesis emulator conditions over their own prosthesis. Nevertheless, these survey results may indicate a benefit of prostheses designed to mitigate disturbances due to uneven terrain.

Table 2. Prosthesis Condition Preferences

Participant	1st	2nd	3rd
1	Cancel, Spring		Prescribed
2	Cancel	Spring	Prescribed
3	Spring	Prescribed	Cancel
4	Cancel	Spring	Prescribed
5	Cancel, Prescribed		Spring

Table 3: replace "Adaptive" with "Cancelling" since that is how it is referred to throughout the rest of the paper. It also might be good to mark each of the conditions with their one letter abbreviation to help connect the first 3 data columns to the last 3. Also, why was the pairwise statistical comparison only done on significant results? I guess it's because from the ANOVA we already know those numbers won't be significant, but as a reader, I still wanted to see the comparison.

We have incorporated these suggestions and updated the table:

Table 3. Group-level Means (and Standard Deviations) of User Outcomes, Normalized to Level Ground Walking

Dependent measure	Prescribed (P)	Spring-like (S)	Cancelling (C)	p-value (ANOVA)	p-value (P/S)	p-value (P/C)	p-value (S/C)
Metabolic rate	1.072 (0.121)	1.156 (0.121)	1.153 (0.082)	0.271	0.161	0.173	0.965
Step width variability	1.690 (0.339)	1.607 (0.354)	1.408 (0.264)	0.045	0.406	0.018	0.069
Step length variability	1.698 (0.427)	1.923 (0.517)	1.970 (0.249)	0.593	0.436	0.352	0.871
Step time variability	1.825 (0.302)	1.808 (0.272)	1.787 (0.318)	0.951	0.889	0.758	0.866
Trunk movement variability	1.289 (0.292)	1.286 (0.279)	1.092 (0.211)	0.026	0.965	0.017	0.018
Mediolateral center of mass motion	0.992 (0.185)	0.979 (0.154)	1.019 (0.157)	0.504	0.710	0.441	0.265

Significant differences ($p < .05$) are in bold.

Page 10

Figure 5: Because all of this data is normalized to the level walking condition, the “1” should be kept at a consistent height on each of the plots or it should be highlighted in some manner. As it is currently, it is hard to compare the 3 conditions to the level ground case. Also, the statistically significant * is a bit hard to understand on the trunk mobility plot. P-S-C are significantly different? P-S are significantly different despite looking the same?

We have updated the figure to indicate $y=1$ for all subplots in the figure for ease of comparison to level ground. Thank you for catching the error in the Trunk Movement Variability plot: S-C should be significant instead of P-S.

Figure 5. User outcomes while walking on uneven terrain for the three prosthesis conditions. Results are normalized to walking on a level treadmill using their prescribed device, indicated by the dashed line. *Statistical significance achieved ($p < 0.05$).

7-14: Biological feet don't change height either. Could we improve the stability of the user by doing something so different from biology? Would that affect the contralateral limb?

It is true that biological feet cannot extend vertically; however, ankle plantarflexion can compensate for lowered portions of the terrain while knee flexion can compensate for raised

portions. This requires a much more complex control scheme than the simple one presented in this manuscript. A prosthesis capable of height adjustment would be a simple approximation of these biological adaptations, but would require knowledge of upcoming terrain. The effect of height compensation on the contralateral limb is very interesting and certainly warrants further research. We have included these points in the main text.

One drawback of the irregularity-cancelling controller is that it is unable to compensate for differences in foot height at each step while walking on the terrain surface. In other words, although the irregularity-cancelling controller significantly reduces the effects of the slope at each step, the absolute height of each step may still differ between feet. Humans can adapt to differences in ground height using a combination of changes in posture at the ankle, knee and hip. However, this requires a much more complex control scheme than the simple one presented in this manuscript. It is possible that these height discrepancies on each step have a large influence on energy cost and kinematic variability when walking over uneven terrain. If this is the case, then the irregularity-cancelling controller is only compensating for a portion of the disturbances. The prosthesis emulator is capable of 1.3 cm of height adjustment, so future studies could compensate for changes in foot height to further mitigate the effects of uneven terrain. However, measuring the absolute position of the prosthesis with sub-centimeter resolution using fieldable sensors will be a technical challenge.

15-29: The prosthesis socket can also be a source of instability that may dominate the results. Was there any thought for the socket fit when selecting subjects for this study?

Although we did not screen participants based on their socket fit, we do not believe this to have affected our study. Movement within a loose-fitting socket would primarily be caused by forces on the prosthesis. If we imagine a user with a particularly loose socket, the compliance due to movement inside the socket would reduce the magnitude of torque disturbances recorded at the prosthesis. In such a case, the torque variability would look similar between the spring-like controller and the irregularity-cancelling controller. However, we know this is not the case since the torque variabilities were significantly different between the two controllers for all five participants. We have updated this paragraph to address this point.

Interactions between the prosthesis emulator and the user are complex and another possible reason for the limited response in user outcomes. Since all the participants had a unilateral amputation, their intact limb was still subject to the disturbances due to uneven terrain. When walking over uneven terrain, people with a lower-limb amputation tend to rely more on their intact limb \cite{gates2012gait,gailey2008,mattes2000}. It is possible that results are dominated by disturbances to the contralateral limb and changes to the affected limb have a smaller influence on energy cost and kinematic variability. Another possibility is that users are affected by reduced sensory feedback on their affected limb. Reduced magnitude or fidelity of sensory feedback could lead to uncertainty about the state of the prosthetic foot and may make it difficult for users to determine if disturbances are being canceled. Socket fit can also affect balance during walking [Samitier 2016], with more compliant sockets leading to larger movements of the prosthesis with respect to the residual limb during large torques produced by uneven terrain.

However, we found that torque variability was significantly reduced with the irregularity-cancelling controller compared to the spring-like controller, suggesting that prosthesis behavior, rather than socket behavior, was dominant in this case.

A fourth possibility is that each participant experiences different benefits from the irregularity-cancelling controller, depending on their priorities during walking. For example, Participant 1 showed reductions in step width and trunk movement variability, whereas Participant 3 only showed reductions in metabolic rate. If individuals indeed experience different benefits, then taking an average across the group washes out individual reductions. In other words, since people with amputation tend to exhibit highly variable gait patterns, it is possible that each participant draws their own benefit from irregularity-cancelling assistance.

47-49: Thanks for mentioning this as a potential source of bias.

Thank you.

Page 12

8-13: Why wouldn't the subjects look down? Were they instructed not to? When walking on rough terrain, humans tend to look down and as the terrain gets rougher they look ahead fewer steps (Matthis 2018). Can we be certain that the subjects are using a natural gait when they are knowingly walking on an unpredictable surface and actively not watching where they are stepping?

This is an excellent point. However, while it is possible to preview your steps during overground walking, it is not the case for this uneven terrain treadmill. Participants were not instructed on where to look but when asked why they chose to look forward, they responded that looking down at the terrain was disorienting and more difficult than simply looking forward. We have edited the main text to include these points:

Participants were not given instructions on how to navigate the terrain or where to look while they walked. Although they were free to look down at the terrain, all participants looked forward during the experiment, and some mentioned that looking down at the terrain was disorienting. However, even if participants opted to look down, the nature of a treadmill makes it impossible to preview upcoming steps, save for the one about to be encountered, and participants would be little able to compensate for upcoming disturbances nor place their foot on easier sections of terrain. This suggests our results may not directly translate to overground terrain, where research has shown that people look ahead to plan their steps [Matthis 2018].

Data:

Looking through your provided code, some files suggest 9 total participants in the study. Is this accurate? And if so, why were the others excluded from the analysis?

Although we have more than 5 participants in our database, not all of these individuals were interested in or eligible for this particular study. Only 5 people participated in this study and no participants were excluded. A readme file has been added to the dataset to improve ease of use and some analysis code has been modified to allow it to run "out of the box".

Reviewer: 2

Comments to the Author(s)

In this paper the authors present a novel controller for an ankle-foot prosthesis designed to help the amputee cope more easily with uneven terrain, as they might experience in daily life. The controller is tested using a prosthetic emulator system, compared to a different controller, and the participants' regular prosthesis. Testing was conducted on an adapted treadmill with an uneven surface atop the treadmill belt. Various outcome measures are used to assess the effects of the controller. The biomechanical outcomes are focussed on the variability of that metric and how it is affected by the prosthetic controller. The work is highly novel, relevant and interesting. The paper is clearly written. The nature of the work strikes me as more exploratory in nature, rather than a complete piece of experimental work, but is still suitable for the remit of this journal. Below are my specific comments.

Thank you for your thoughtful comments.

As stated above the work is somewhat exploratory, and therefore I think the title of the paper is too firmly stated. While I appreciate that the nature of this work does not lend itself to large cohorts, with only 5 participants, each of which is likely to have an individual response to the intervention, the study lacks the power to draw such firm conclusions. This compounded by the other limitations that the authors acknowledge, such as a limited familiarisation to the intervention. The title suggests that a clear result has been reached, and that the study disproves any value in the control strategy. The paper itself does not carry this tone, and I believe the title should be changed to reflect more closely the nature of the work/findings.

We acknowledge the title may have appeared too firm and have modified it accordingly:

The effects of ground-irregularity-cancelling prosthesis control on balance over uneven surfaces

It would improve the paper if the authors could justify their main outcome metrics in the methods. Intuitively, one might think that reduced variability is a sign of improved stability, but this is not always the case in the execution of motor tasks. It could be that it is not about preventing the variability induced by the terrain, but the individual's ability to respond to it. Therefore, is variability relative to level ground the best metric to assess device efficacy? Some justification in the paper would be beneficial.

Determining the appropriate outcome metrics, especially for experiments studying balance, is a difficult and contentious task; finding a true measure of an individual's level of balance while walking is still an open problem. Papers that study balance often report related measures such as metabolic rate (to determine balance-related effort) or correlates such as step width variability. For this manuscript, we chose to use the same outcome measures reported by other studies involving people with a lower limb amputation walking over an uneven surface. We have edited the text to reflect these points.

Validating a controller designed to improve balance requires measurements of the user's balance. However, finding a true measure of an individual's level of balance while walking is still

an open problem. In this manuscript, we report kinematic outcomes that have previously been shown to be significantly different between people with and without transtibial amputation when walking over uneven surfaces: step width variability [Beurskens 2014, Gates 2013, Sinitski 2019], step length variability [Sinitski 2019], step time variability [Sinitski 2019], trunk movement variability [Beurskens 2014], and mediolateral center of mass motion [Gates 2013]. Additionally, we report metabolic rate, as it can be indicative of balance-related effort [Voloshina 2013, Houdijk 2009, Wezenberg 2011], and also perceived comfort, as it is one of the largest restrictions of prosthesis use [Dillingham 2001].

What were the masses of the different prosthetics and the emulator attachment used in the study? Mass could easily have a significant impact on metabolic power during walking, so this should be included and discussed briefly.

We agree that prosthesis mass can influence metabolic power during walking. Although we did not collect this data during the experiment, we have updated the text to include approximations in the Discussion section:

There are no trends in user outcomes that correlate with the masses of the participants' prescribed prostheses. Although we did not collect the masses of the devices used by the participants, we have provided estimates based on manufacturer data. We supplied shoes for the participants to wear during the experiment, with a mass of 250 g per shoe. The devices used by Participants 1 and 5 (Ossur Pro-flex LP and College Park Truststep) had a mass of about 650 g for the device and footshell. Adding the mass of the shoe results in a sum of 900 g. The devices used by Participants 2 and 3 (Ottobock Triton Harmony Ossur Pro-flex Pivot) had a mass of around 1150 g including footshell and shoe. The Ottobock Empower used by Participant 4 had a mass of around 2450 g including footshell and shoe. For comparison, the prosthesis emulator has a mass of about 1200 g and did not incorporate a footshell or shoe. Although Participant 5 reported that their prescribed foot felt lighter than the prosthesis emulator, we did not observe any trends in user outcomes that favored lighter devices across any of our participants.

Several control parameters were fixed based on the 'user preference'. It would be useful to include a brief description of how user preference was established and what some of these parameters were (maybe in an appendix). Particularly, the threshold angle at which the device switched to 'stance' control seems relevant.

A brief description of how the non-variable control parameters were set have been added to the Methods section and a table of these parameters have been added to Supplementary Text.

Non-variable parameters for the irregularity-cancelling controller, including heel and forefoot stiffnesses, heel neutral angle, plantarflexion velocity, and IMU threshold angle, were all set based on user preference during a fitting session prior to testing. Heel and forefoot stiffnesses, as well as heel neutral angle, were set with guidance from the prosthetist. Plantarflexion velocity was set such that the forefoot digits would contact the ground before the shank angle crossed the threshold value. The IMU threshold angle was set by having the participant walk on a level

treadmill and compare the spring-like and irregularity-cancelling controllers. The threshold angle was adjusted until the participant reported the irregularity-cancelling controller felt identical to the spring-like controller. Adjusting the IMU threshold angle is similar to changing the alignment of a prosthesis to be slightly plantarflexed or dorsiflexed. A table of these parameters can be found in the Supplementary Text.

Supplementary Text:

Table 1 shows a list of prosthesis parameters set during the fitting session prior to the experiment. These parameters were kept constant throughout the experiment.

Table 1: Non-Variable Control Parameters

Participant	Forefoot Stiffness (Nm/rad)	Heel Stiffness (Nm/rad)	Heel Neutral Angle (rad)	Plantarflexion Velocity (rad/s)	IMU Threshold Angle (deg)
1	300	250	2.1	20	0
2	300	200	1.9	20	-6.5
3	300	450	2.1	15	-4
4	200	175	1.9	25	0
5	250	300	1.9	20	-6

Some discussion of the results in comparison to how non-amputees respond to similar uneven terrain situations seems relevant. Results are presented relative to level ground, so are they similar adaptations to those made by non-amputees? (e.g. is an increase in step width expected?). Also in the context of the control scheme – do non-amputees seek to minimise ankle torque variability on uneven terrain, or is a varied response required to account for different perturbations?

This is an important comparison point and we have added it to the Discussion section. A brief discussion on the control scheme individuals without amputation use when walking on uneven terrain is included in the second paragraph in the Introduction section (Page 3 Line 9) and discussed in further detail in the cited paper [Schultz 2018].

Since all measures were normalized to walking on a level treadmill while using their prescribed device, we can use data from the prescribed device condition to quantify how participants adapted their gait in response to uneven terrain (Table 3). Specifically, we can see that terrain had a small or negligible effect on mediolateral center of mass motion and metabolic rate (0.8% reduction and 7.2% increase, respectively) as well as mean step width, length, and time (2.7% reduction, 0.5% reduction, and 2.7% increase, respectively). However, there was a large effect on step width, step length, and step time variabilities (69%, 70%, and 83% increase, respectively). This finding is consistent with previously reported data of individuals with amputation walking on terrain [Gates 2012, Voloshina 2020], and implies participants adapted to the terrain using foot placement strategies and relied less on center of mass or torso

adjustments. These adaptations are also similar to individuals without amputation, who experienced little to no changes in mean step parameters, but experienced significant increases in step width, length, and time variability (35%, 23% and 27%, respectively) [Voloshina 2013]. Notably, the increases in step variability were smaller and the increase in metabolic rate was greater than was reported in this study.

I presume treadmill speed was the same for level ground and uneven terrain. Maybe the natural response would be to walk more slowly on uneven ground?

It is true that people naturally tend to walk more slowly on uneven terrain; however, we wanted to hold participant walking speeds constant in order to maintain a fair comparison between level ground and uneven terrain conditions. We have clarified this point in the Methods Section:

Five healthy adults with unilateral transtibial amputation participated in the study, with an average age, height, and body mass of 47 +/- 17 years, 170 +/- 4.6 cm, and 66 +/- 6.5 kg, respectively (Table 1). Prior to testing, participants were scheduled for a fitting session in which a prosthetist fit them to the prosthesis emulator. Participants then acclimated to walking in the emulator on both level ground and uneven terrain. At the end of the session, we asked the participant to identify their fastest comfortable walking speed while walking on the uneven terrain. This speed was held constant for that participant for all trials of the experiment.